# Comprehensive Co-Inhibitory Receptor (Co-IR) Expression on T Cells and Soluble Proteins in Rheumatoid Arthritis

**DOI:** 10.3390/cells13050403

**Published:** 2024-02-26

**Authors:** Chin-Man Wang, Yeong-Jian Jan Wu, Li-Yu Huang, Jian-Wen Zheng, Ji-Yih Chen

**Affiliations:** 1Department of Rehabilitation, Chang Gung Memorial Hospital, Chang Gung University College of Medicine, Taoyuan 33302, Taiwan; cmw1314@adm.cgmh.org.tw; 2Division of Allergy, Immunology and Rheumatology, Department of Medicine, Chang Gung Memorial Hospital, Chang Gung University College of Medicine, No. 5, Fu-Shin St. Kwei-Shan, Taoyuan 33305, Taiwan; yjwu1962@cgmh.org.tw (Y.-J.J.W.);

**Keywords:** RA, T cell, co-inhibitory receptors, disease activity

## Abstract

Co-inhibitory receptors (Co-IRs) are essential in controlling the progression of immunopathology in rheumatoid arthritis (RA) by limiting T cell activation. The objective of this investigation was to determine the phenotypic expression of Co-IR T cells and to assess the levels of serum soluble PD-1, PDL-2, and TIM3 in Taiwanese RA patients. Methods: Co-IRs T cells were immunophenotyped employing multicolor flow cytometry, and ELISA was utilized for measuring soluble PD-1, PDL-2, and TIM3. Correlations have been detected across the percentage of T cells expressing Co-IRs (MFI) and different indicators in the blood, including ESR, high-sensitivity CRP (hsCRP), 28 joint disease activity scores (DAS28), and soluble PD-1/PDL-2/TIM3. Results: In RA patients, we recognized elevated levels of PD-1 (CD279), CTLA-4, and TIGIT in CD4+ T cells; TIGIT, HLA-DR, TIM3, and LAG3 in CD8+ T cells; and CD8+CD279+TIM3+, CD8+HLA-DR+CD38+ T cells. The following tests were revealed to be correlated with hsCRP: CD4/CD279 MFI, CD4/CD279%, CD4/TIM3%, CD8/TIM3%, CD8/TIM3 MFI, CD8/LAG3%, and CD8+HLA-DR+CD38+%. CD8/LAG3 and CD8/TIM3 MFIs are linked to ESR. DAS28-ESR and DAS28-CRP exhibited relationships with CD4/CD127 MFI, CD8/CD279%, and CD8/CD127 MFI, respectively. CD4+CD279+TIM3+% was correlated with DAS28-ESR (*p* = 0.0084, N = 46), DAS28-CRP (*p* = 0.007, N = 47), and hsCRP (*p* = 0.002, N = 56), respectively. In the serum of patients with RA, levels of soluble PD-1, PDL-2, and Tim3 were extremely elevated. CD4+ TIM3+% (*p* = 0.0089, N = 46) and CD8+ TIM3+% (*p* = 0.0305, N = 46) were correlated with sTIM3 levels; sPD1 levels were correlated with CD4+CD279+% (*p* < 0.0001, N = 31) and CD3+CD279+% (*p* = 0.0084, N = 30). Conclusions: Co-IR expressions on CD4+ and CD8+ T cells, as well as soluble PD-1, PDL-2, and TIM3 levels, could function as indicators of disease activity and potentially play crucial roles in the pathogenesis of RA.

## 1. Introduction

Rheumatoid arthritis (RA) is a widely observed autoimmune disorder distinguished by the gradual deterioration of articular structures, inflammation of the joints, and alterations in the cardiovascular and metabolic systems. The pathogenesis of RA is predominantly attributed to an autoimmune dysfunction that progresses through conceptual phases. Gaining an understanding of the underlying causes that instigate and sustain the disease represents a potential avenue for advancement [1]. Individuals diagnosed with RA have demonstrated that their central immune system gets triggered by innate myeloid/lymphoid cells, promptly defensive natural killer (NK) cells, mast cells, and adaptive T and B cells [2,3]. Progressive joint degeneration ensues as a consequence of the anomalous cytokine production and cell proliferation initiated through these immunological encounters [2,3,4]. Exceptionally robust immune treatment options encompass achieving the development of tolerance and regaining of immunologic homeostasis with the goal of attaining drug-free remission [5].

Therapeutic interventions such as T cell co-stimulation signaling blockades, inhibition of pro-inflammatory cytokines (Tumor Necrosis factor-α (TNFα) and Interleukin-6 (IL-6)), and B cell depletion, reveal that RA is a disorder that has numerous root causes. Immunoreactivity modulation is impacted by the complicated interaction of a multitude of molecules within cells. Co-inhibition, an immunologic phenomenon, is increasingly recognized as a potential development target for immune-mediated diseases. Lackluster co-inhibition and/or overbearing co-stimulation may contribute to the emergence of autoimmunity. Investigating these underlying processes provides the potential for discovering innovative therapy targets for treatment-resistant RA patients. T cell exhaustion is distinguished by a substantial decrease in T cell proliferation and the loss of effector functionality [6]. Standardized assays for immunophenotyping that determine RA clinical profiles enable novel prospects for patient stratification and prioritize therapeutic adaptations [7].

The immune system possesses a unique ability to safeguard the body and ensure homeostasis via the functioning of multiple balances and checks. T cell receptors must exist for T cell activation, and cytokines generate yet another co-stimulatory signal through an immune co-receptor which has been enhanced [8]. Co-inhibitory receptors (Co-IRs), which play dynamic roles in immunological homeostasis and autoimmunity tolerance modulation, are expressed by a wide range of immune cells [9,10]. Cytotoxic T lymphocyte antigen 4 (CTLA4), programmed cell death 1 (PD-1), lymphocyte activation gene 3 (LAG3), T cell immunoglobulin and mucin domain-containing 3 (TIM3), and T cell immunoglobulin and ITIM domain (TIGIT) all play important roles in immune responses and T cell homeostasis [11]. Binding to PD-1, PD-L1 (programmed cell death 1 ligand 1), or PD-L2 (programmed cell death 1 ligand 2) promotes subsequent PD-1-linked transduction events. This can impair TCR signaling via feedback interference and diminish anti-apoptotic proteins, thereby limiting T cell survival, proliferation, and immunological function. An immune checkpoint modulates immune activation and self-tolerance [12,13]. Strengthening IR signals (agonism), primarily CTLA4, PD1, LAG3, TIM3, and TIGIT, represents an unexplored and underappreciated therapeutic potential [14]. In a murine model, TIGIT engagement had been proven to ameliorate autoimmunity, and pathway modification is used to generate durable tolerance for the treatment of autoimmune conditions [15]. Interleukin-7 receptor-α (CD127) and common-γ chain receptor-formed heterodimer complex-driven IL-7 signaling is vital for naive T cell survival, additionally to the growth and survival of memory T cells, and exerts an impact on immune-mediated autoimmune diseases such as RA [16]. CD244 (SLAM 4; signaling lymphocytic activating molecule 4) binds to the ligand CD48 on cells nearby and conveys stimulatory or inhibitory signals to immune response cells, contributing to the emergence and advancement of autoimmune disorders [17].

Co-IRs have a vital role in controlling the continuing immune-mediated inflammation in RA. Their ability to reduce T cell activation indicates that stimulating these pathways of inhibition on a personal or systemic magnitude could serve as an effective approach to establish biomarkers of RA activity. The aims of this study were to ascertain the immunological phenotypic expression of Co-IRs on T cells and soluble mediators that correspond with disease activity in patients with RA in Taiwan in accordance with this theory.

## 2. Patients and Methods

### 2.1. Descriptive Attributes of Cohorts and Study Populations

The majority of the participants in this study were enrolled in the rheumatology clinics affiliated with Chang Gung Memorial Hospital. All the RA patients enrolled in the study fulfilled the criteria set forth by the American College of Rheumatology/European League Against Rheumatism [18]. Appendix A summarizes the characteristics of 48 RA patients (age: 58.4 ± 13.5 years; gender: male/female, 10/38) and 27 healthy controls (age: 36.3 ± 5.7 years; gender: male/female, 9/18). Among RA patients, 32 were rheumatoid factor (RF)-positive (>15 IU/mL), with 21 having high titers (>100 IU/mL). The RA X-ray stage distribution was 11 in I, 7 in II, 14 in III, and 16 in IV. Conventional Disease-modifying antirheumatic drugs (DMARDs) used include methotrexate (27), sulfasalazine (28), hydroxychloroquine (38), and leflunomide (10). Biologic agents included five TNF-blockers and one IL-6 receptor blocker (Actemra, Chugai pharmaceutical, Tokyo, Japan), as well as eight targeted synthetic JAK inhibitors (Tofacitinib). The human research protocols received approval from the ethics committee at Chang Gung Memorial Hospital (Institutional Review Board No. 201800457B0). Prior to conducting any measurements, interventions, or blood collections, informed consent was obtained from all human subjects.

### 2.2. Cell Isolation

Peripheral blood mononuclear cells (PBMCs) mononuclear cells (SFMCs) were directly isolated by Ficoll–Hypaque gradient (Sigma-Aldrich, Burlington, MA, USA).

### 2.3. Antibodies for Immune-Phenotype

CD3-PerCP-Cy5.5, CD4-PerCP-Cy5.5, CD8-APC, CD38-PE, CD127-FITC, CD127-PerCP-Cy5.5, CD160-Alex Fluor 488, LAG3(CD223)-PE, CD244-PE, PD-1(CD279)-FITC, CTLA-4-Alex Fluor 488, HLA-DR-FITC, TIGIT-PE, and TIM3-PE antibodies were included.

### 2.4. Flow Cytometric Analysis

All patients who initiated treatment with conventional DMARDs were enrolled in an immune-phenotyping study using multicolor flow cytometry. Blood was drawn as part of standard laboratory testing procedures. Flow cytometry was used to promptly analyze all of the obtained samples. To inhibit Fc receptors and prevent non-specific antibody binding, peripheral blood mononuclear cells were extracted and suspended in PBS containing 3% human IgG. They were then incubated in the dark for 15 min at 4 °C. Following this, the cells were rinsed with PBS containing 1% BSA. The background fluorescence was evaluated utilizing isotype- and fluorochrome-matched control mAbs. Antibodies were used to stain cells, and FACSCanto II multicolor flow cytometry was used to evaluate the results. The cells were collected and analyzed using FlowJo, LLC software (version 10.5.3, Tree Star, Ashland, OR, USA). The phenotype of immune cell subsets was determined utilizing a full four-color flow cytometric analysis adopting the HIP method of standardized immune-phenotyping procedures, as well as a detailed understanding of the parameters of healthy vs. diseased or perturbed human immune systems. Clinical disease activity was correlated with the results, which included high-sensitivity CRP (hsCRP), ESR, and 28 joint disease activity scores (DAS28-CRP and DAS28-ESR).

### 2.5. Enzyme-Linked Immunosorbent Assay (ELISA)

The serum levels of specific related Co-IRs and ligands for soluble PD-1, PDL-2, and TIM3 analysis were measured using ELISA kits (R&D Systems, Minneapolis, MN, USA), according to the manufacturer’s instructions.

### 2.6. Statistical Analysis

The statistical analyses were performed with Graph Pad Prism 8.4.1. Data are presented as mean plus or minus standard deviation (S.D.) or percentage (%). The differences between groups were examined using an unpaired *t*-test. Data for relatively small populations are expressed as median and interquartile range. Differences between groups were compared using non-parametric Mann–Whitney U testing. Correlation analysis was performed using Pearson’s correlation coefficient. By using Spearman correlation analysis, the relationship between Co-IR expression/soluble levels and DAS28-CRP and DAS28-ESR was evaluated. The *p*-value for significance was adjusted for multiple testing using the Bonferroni adjustments.

## 3. Results

### 3.1. Co-IR Expression on T Cell Characterization with Taiwanese RA

Flow cytometry was utilized to determine the percentages (%) and mean fluorescence intensities (MFI) of Co-IRs on CD4+ and CD8+ T cells isolated from peripheral blood. Significantly higher percentages and absolute quantities of several Co-IRs were expressed in patients with RA compared to the control group. PD-1 (CD279) expression percentages on CD4+ T cells was significantly greater on RA patients’ CD4+ T cells instead of on those from healthy volunteers, as shown in Table 1 (4.6 (9.625), (N = 48) vs. 3.1 (6.9), (N = 27), *p* = 0.0088). Additionally, RA patients showed higher TIGIT expression percentages on CD4+ T cells relative to healthy volunteers (25.35 (11.23), (N = 48) vs. 20.1 (5.73), (N = 26), *p* = 0.0016). CD127 expression percentages and MFI was significantly reduced on CD8+ T cells of RA patients compared to healthy volunteers (21.2 (22.63), (N = 48) vs. 40.2 (36.2), (N = 27), *p* = 0.0012; MFI: 352 (92), (N = 33) vs. 402 (102), (N = 25), *p* = 0.0003). RA patients exhibited significantly higher TIGIT expression percentages and greater MFI on CD8+ T cells relative to healthy volunteers (47.95 (23.65), (N = 48) vs. 25.4 (12.9), (N = 27), *p* < 0.0001; MFI: 662 (187), (N = 33) vs. 567 (120.5), (N = 25), *p =* 0.0183). Compared to healthy volunteers, RA patients exhibited higher HLA-DR expression percentages and MFI on CD8+ T cells (43.8 (24.8), (N = 31) vs. 15.4 (13.85), (N = 17), *p* < 0.0001; MFI: 672 (265), (N = 31) vs. 537 (100), (N = 17), *p* = 0.0013). In contrast to healthy volunteers, TIM3 expression percentages and MFI on CD8+ T cells proved significantly greater in RA patients (12.9 (20.255), (N = 48) vs. 7.5 (8.8), (N = 27), *p* = 0.0012; MFI: 457 (123.8), (N = 48) vs. 395 (106), (N = 27), *p* = 0.0001). In RA patients, the proportion (%) of LAG-3 CD8+T cells was substantially greater compared to those of healthy volunteers (0.55 (1), (N = 46) vs. 0.1 (0.3), (N = 27), *p* = 0.0015). The study identified the following percentages of total T cells expressing CD3: CD3+CD279+ (4.7 (9.33), (N = 46) vs. 2 (5.7), (N = 27), *p* = 0.025) and CD3+TIGIT+ (40 (31.8), (N = 47) vs. 22.5 (18.6), (N = 27), *p* = 0.0001) were expanded in RA patients compared to healthy volunteers. T cells expressing CD8+CD279+TIM3+ (0.2 (0.325), (N = 46) compared to 0.1 (0.1), (N = 26), *p* = 0.0052) and CD8+HLA-DR+CD38+ (7.5 (6.6), (N = 47) compared to 3.1 (3.3), (N = 27), *p* = 0.0004) were also significantly more prevalent in RA patients than in healthy volunteers (Table 2).

### 3.2. Correlation between RA Disease Activity and Co-IRs Expression on T Cells

Co-IR expression regulates the continuous inflammatory process; therefore, disease activity in RA is assessed by monitoring Co-IR levels. The disease activity of fifty-seven RA patients was evaluated, nine of whom had been determined repeatedly. As can be seen in Figure 1A,C, CD4/PD1 MFI in CD4 T cells were linked with hsCRP (*p* = 0.0393, N = 56) and ESR (*p* = 0.0096, N = 51). There was a correlation between hsCRP and CD4/CD279% (*p* = 0.0251, N = 56, Figure 1B), CD4/TIM3% (*p* = 0.0459, N = 56, Figure 1D), and CD4/TIGIT MFI (*p* = 0.0156, N = 45, Figure 1E). CD4/CD127 MFI were correlated to DAS28-ESR (*p* = 0.0321, N = 40, Figure 1F) and DAS28-CRP (*p* = 0.0145, N = 40, Figure 1G). As shown in Figure 2A,B, CD8/TIM3 MFI (*p* = 0.0459, N = 51) and CD8/LAG3 MFI (*p* = 0.0182, N = 39) were correlated to ESR. There was a correlation between hsCRP and CD8/TIM3% (*p* = 0.0114, N = 56, Figure 2C), CD8/TIM3 MFI (*p* = 0.001, N = 56, Figure 2D), CD8/LAG3% (*p* = 0.0236, N = 56, Figure 2E), and CD8+HLA-DR MFI (*p* = 0.0128, N = 43, Figure 2F). DAS28-ESR was linked with CD8/CD279% (*p* = 0.0456, N = 46, Figure 2G) and CD8/CD127 MFI (*p* = 0.0063, N = 40, Figure 2H). Additionally associated with DAS28-CRP were CD8/CD279% (*p* = 0.0363, N = 47, Figure 2I) and CD8/CD127 MFI (*p* = 0.0157, N = 40, Figure 2J). ESR and CD3/CD244% showed a correlation (*p* = 0.0405, N = 51, Figure 2K). As shown in Figure 3, CD4+CD279+TIM3+% was linked with hsCRP (*p* = 0.002, N = 56, Figure 3A), DAS28-ESR (*p* = 0.0084, N = 46, Figure 3B), and DAS-28 CRP (*p* = 0.002, N = 47, Figure 3C), CD8+CD127+TIGIT+% were correlated to ESR (*p* = 0.0355, N = 51, Figure 3D), CD8+HLA-DR+CD38+% with hsCRP (*p* = 0.0243, N = 55, Figure 3E) and CD3+CD160+CD244+% were correlated with ESR (*p* = 0.0234, N = 51, Figure 3F). Appendix A show the detailed relationships between the activity of the RA disease and the expression of Co-IRs on T cell activity.

### 3.3. Soluble PD-1, PDL-2, and TIM3 Levels in RA and Correlation with Cell Expression

As shown in Figure 4A–C, RA patients have significantly elevated serum levels of soluble PD-1, PDL-2, and Tim3 compared to normal controls (sPD1: 450.5 ± 89.01, *n* = 31 vs. 111.1 ± 32.92, *n* = 92 *p* < 0.0001; sPD-L2: 17.9 ± 1.566, *n* = 47 vs. 12.78 ± 0.2314, *n* = 92 *p* < 0.0001; sTIM3: 4.288 ± 0.3015, *n* = 40 vs. 3.675 ± 0.08441, *n* = 92 *p* = 0.0104). sTIM3 levels were correlated with CD4+ TIM3+% (*p* = 0.0089, N = 46, Figure 4D) and CD8+ TIM3+% (*p* = 0.0305, N = 46, Figure 4E). sPD-1 levels were correlated to CD4+CD279+% (*p* < 0.0001, N = 31, Figure 4F). Our data indicate that soluble PD-1, PDL-2, and TIM3 levels can also serve as serologic markers of disease.

## 4. Discussion

The findings of this study illustrate the fact that augmented expression of Co-IRs (PD-1, CTLA-4, TIGIT, and TIM3) in CD4+ and CD8+ T cells is crucial for the emergence and progression of RA. PD-1+ T cells are detectable in the RA synovium and synovial fluids [19], and there is an upward correlation with RA activity [20,21]. PD-1 is widely distributed across myriad T cell subsets and is responsible for the differentiation of a variety of distinctly functioning T cells. In the phase 2a trial, peresolimab, a humanized IgG1 monoclonal antibody which stimulates the PD-1 pathway, was found to be more effective in RA patients, with DAS28-CRP reductions from baseline being considerably greater in the 700 mg peresolimab group [22]. The CTLA-4 Ig fusion molecule blocks the co-stimulation of T cells, and is frequently anticipated by utilizing T follicular helper cells (Tfh) that are CD4+PD-1+CXCR5+ at baseline [23] and over a range of chronic autoimmune inflammatory disorders [24,25,26]. Extra-lymphoid inflammatory tissue, including the synovium of patients with RA, contains PD-1^hi^CXCR5-CD4+ peripheral helper T (Tph) cells. These cells are known to drive plasmablast responses and generate antibodies, implying that they could serve as a viable target for therapy [27,28]. Tph cells are indicative of the severity of RA [29]. Seropositive early RA patients with active disease exhibit elevated Tph cells, whereas Tfh cells are constitutively elevated [30].

Tfr (T follicular regulatory; CD4+CXCR5+Foxp3+) and Tfh cells are both crucial for humoral immunity responses; the Tfr/Tfh ratio of cells has been associated with the onset of RA [31,32]. PD-1+ICOS+ Tfh, Tfh17-like, and Tfh1/17-like cells perform opposing tasks to Tfr-like and mTfr-like regulatory cells [22,23] which contribute to cytokine disequilibrium in the advancement of RA [33]. CD8+ T cells are starting to assume an integral part in RA; PD-1^high^CD8+ T cells, which are akin to CD4 Tph T cells, have the capacity of emitting IL-21 present plasmablast auxiliary functions [34]. In RA, smoking reactivates exhausted CD8+ T cells to perform effector roles through the substitution of cytotoxic CD107 for PD-1 and in accordance with serum survivin values [35]. It is worth noting that PD-1 may exhibit distinct involvement in auto-inflammatory reactions in psoriatic arthritis (PsA) and autoimmune processes in RA [36]. A correlation is being observed between the presence of PD-1-expressing CD4+ and CD8+ T cells and the extent of disease in Taiwanese patients with RA.

Song et al. observed that the inhibitory effect of TIM3 and IL-37 on inflammation could possess an aspect in the development of RA. Nonetheless, no correlation was noticed between DAS28 and TIM3 expression on diverse T cells [37]. In contrast, a negative correlation has been noted by others across the levels of Tim3 on circulating CD4+ and CD8+ T cells and DAS28 in RA patients [21,38,39]. TIM3 and PD-1 co-expression possesses disease-suppressive properties in RA [40]. In addition, PD-1 and TIM3 expression were found to be enhanced on Tph entity cells in bronchoalveolar lavage fluid from patients with interstitial lung disease (ILD). Expression of Co-IRs in peripheral blood distinguished idiopathic ILD from that of RA [41]. We identified that CD4+CD279+TIM3+ cells exhibited a correlation with DAS28-CRP, hsCRP, and DAS28-ESR, all of which could potentially function as biomarkers.

In this study, CD4+ and CD8+ T cells of RA patients evidenced TIGIT upregulation. Positive correlations have been noted across TIGIT expression levels and frequency on CD4+ and CD8+ T cells and inflammatory markers (CRP and ESR), autoantibodies (Anti-CCP (anti-cyclic citrullinated peptides) and RF), and the DAS28 score in RA [42]. As a therapeutic target, inhibition of the TIGIT pathway induces a variety of autoimmune diseases, while TIGIT function augmentation ameliorates autoimmune settings in mice [15]. In mice, TIGIT introduction at the onset of the disease more effectively diminished anti-collagen II antibody levels and disease severity [43]. The true populations of Treg cells, CD4+CD25^hi^CD127^low^Foxp3+Helios+ T cells, were reduced and inversely correlated with disease activity in RA patients. On the contrary, CD226 and TIGIT expression levels were increased in CD4+Foxp3+ cells, yet they were inconsistent with RA activity or progression [44]. The co-expression of TIGIT and PD-1 on CD4+T cells has been linked to autoantibody production and RA activity [45]. First-degree relatives who are autoantibody-positive have a higher prevalence of TIGIT+ CD4 T cells [46]. Consequently, TIGIT expression on CD4 T cells is crucial for the production of autoantibodies by B cells in RA.

Significantly increased CD8+CD279+TIM3+T cells, which might be exhausted T (T_EX_) cells, were observed in the RA patients included in this study. Persistent antigen exposure causes T cell exhaustion, which in turn manifests in a hierarchical and gradual decline in effector function owing to overstimulation. As a consequence, the cells fail to eliminate an infection or tumor. Notwithstanding their compromised functionality, T_EX_ cells still manage to impart a certain degree of control over persistent viral infections and impede tumor growth [47]. Calcium flux and calcineurin signaling, which enable epigenetic modifications that enforce the exhausted state, are utilized to induce TOX in response to persistent antigen exposure [48]. An exhausted phenotype is induced in CD8+ T cells through TOX interactions with nuclear receptor subfamily 4 group A (NR4A) when coupled with an elaborate T cell signaling network [49]. The distinguishing characteristic of T_EX_ cells is their co-expression of TIGIT, LAG3, PD-1, CTLA4, and TIM3. Extracellular vesicles (EVs) exert a crucial part in the development of RA, carrying microRNAs (miRNAs) corresponding to PD-1 along with additional indications of T cell suppression, which may lead to T cell exhaustion [50]. EVs have been demonstrated to promote the production of inflammatory mediators, including chemokines and cytokines [51]. Co-IR enhancement resulting in T cell exhaustion could imply that distinct T cell differentiation functions improperly at various stages of RA.

CD38 catalyzes the metabolism of nicotinamide adenine dinucleotide (NAD); thus, NAD deprivation is a prominent pathological factor [52]. CD38^high^ NK cells, as opposed to CD38^low^ NK-like T cells, disrupt immunity, impeding the differentiation of Treg cells through the activation of mTOR signaling in CD4+ T cells [53]. HLA-DR+CD38^high^ CD8+ T cells demonstrated the greatest degree of hyperactivation and were the most vigorously primed [54]. Significantly prevalent HLA-DR+CD38+CD8+ T cells were identified in Taiwanese RA patients, suggesting that CD8 T cell hyperactivation plays a crucial role in RA pathogenesis. Activated CD38^hi^ CD8 T cells are detected in the T cell infiltrate of synovial tissue generated by inflammatory arthritis following PD-1 block therapy [55]. Persistent sufferers of hepatitis C are bombarded by heavily cytotoxic CD38+HLA-DR+CD8+ T cells, which elicit liver damage [56]. The survival or mortality of H7N9 patients can be distinguished based on clonal expansion kinetics [57]. Hyperactivation and dysregulation of HLA-DR+CD38^high^CD8+ T cells result in exhaustion within COVID-19 patients with severe disease [58]. The survivors’ proportion of CD8+CD38+ cells declined subsequent to their recovery from COVID-19 [59]. Acute graft versus host disease (GVHD) can be recognized by a profusion of proliferating, activated CD38^high^CD8+ T cells that are cytotoxic in nature. These cells lack the capacity to react to the reactivation of CMV or EBV [60]. The correlation between CD8+HLA-DR+CD38+% and hsCRP implies a causal relationship with persistent RA disease activity.

TIM3 pathways and ligands that connect PD-1 and PD-L1/2 might mitigate RA autoimmunity. sTIM3 and sPD-1 can compete with the ligand, thereby blocking the inhibiting nature of membrane-bound PD-1 and TIM3 signals and facilitating T cell exertion [61]. sPD-1 accelerates the onset of collagen-induced arthritis through Th1 and Th17 pathways [62]. sTIM3 correlated significantly with ESR and MMP-3 in RA patients with moderate ACPA (anti-citrullinated protein autoantibodies) titers (200 U/mL) [63]. Elevated plasma levels of sTIM3 are indicative of disease activity in the early stages of RA [40]. sPD-1 is a key mediator for predicting the development of inflammatory and radiographic progression in early RA [64]. sPD-1 and sPD-L1 surges may serve as useful biomarkers for identifying the onset of ILD in patients with RA [65,66]. Divergent control of sPD-L1 during the early and late stages of RA may be indicative of an evolution from acute to chronic inflammation [67]. CD4 T cell resistance to PD-1-mediated suppression is increased in RA and psoriatic arthritis (PsA), which may be partially due to the presence of sPD-1 in the inflammatory milieu [68]. We established a correlation between elevated levels of soluble PD-1, PDL-2, and TIM-3 in RA and disease activity as a biomarker.

As immune-related adverse events (irAEs) emerge with the administration of immune checkpoint inhibitors (ICIs), inflammatory syndromes are evolving to encompass distal tissue injury and autoimmune disorders, which impact nearly all organ systems [69]. These processes are critical to the proper functioning of T cells, whereas cancer immunotherapies boost T cells by blocking negative signals on T lymphocytes, resulting in abnormalities in the co-inhibitory pathway that may be contributing to rheumatic irAEs [70]. Blocking PD-1 and CTLA4, along with augmenting antitumor immunity, motivates the creation of self-reactive T cell-induced inflammatory arthritis with increased Th17 and transitory Th1/Th17 cell characteristics [71]. An extensive spectrum of musculoskeletal irAEs that do not meet the conventional diagnostic criteria of rheumatic illnesses necessitate thorough investigation [72]. It is crucial to advance forecast rheumatic irAE onset [73] and tailor therapy that does not affect the anti-tumor efficacy of ICIs [74].

The major concern of this study is the precise identification of T cell immunophenotype contributions to the pathogenesis of RA, which single-cell RNA sequencing (scRNA-seq) may be able to resolve. Therefore, additional research is required to identify the molecular properties. RA-causing cells are not found in the peripheral circulation, where they operate to instigate the inflammatory response; rather, they reside primarily in the synovium. As a result, a wide range of investigations concerning T cells isolated from synovial biopsy samples are presently in progress. For further investigation of particular target pathways, larger samples exhibiting a range of biologic agent responses are necessary.

## 5. Conclusions

This study is valuable in the clinical setting for identifying potential markers with crucial roles in the progression and monitoring of RA disease activity. Our study unveils the biological processes by which Co-IR T cells dampen the immune system broadly while offering a justification for the use of soluble PD-L1 as a predictor of RA disease activity and therapeutic response. Our novel findings may increase the likelihood of individualized treatment and suggest new possible avenues of therapy for RA.

## Figures and Tables

**Figure 1 cells-13-00403-f001:**
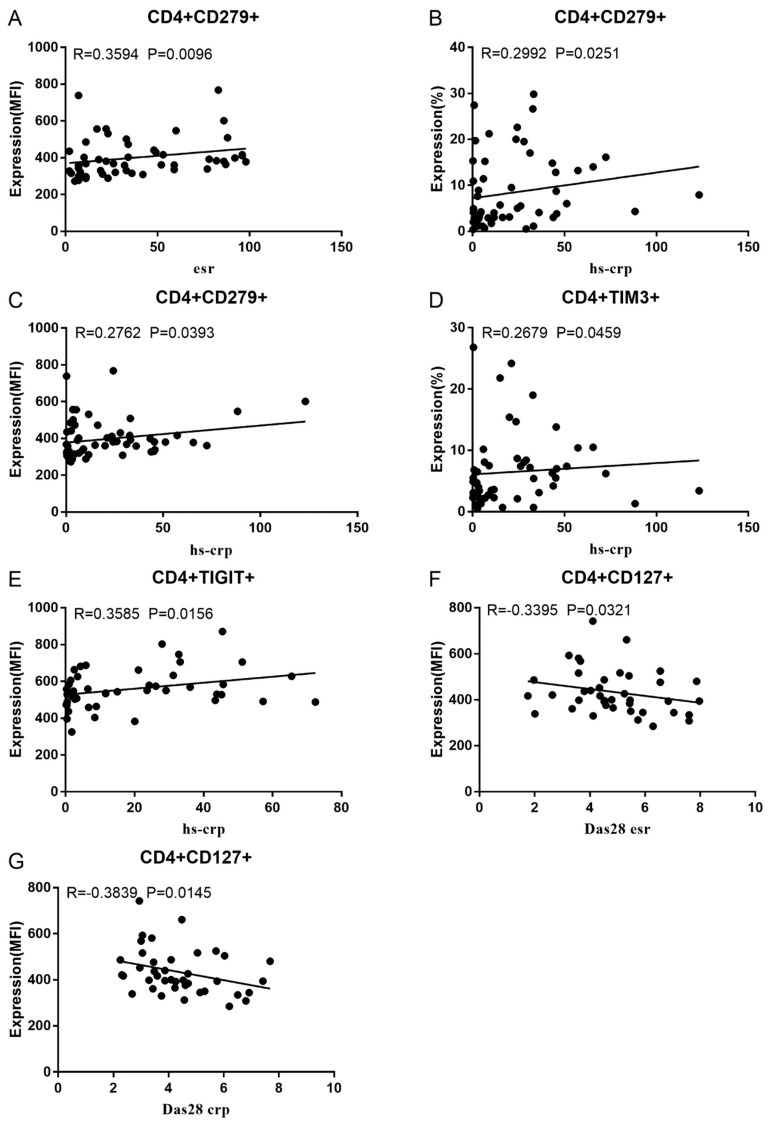
Significant correlation between RA disease activity and Co-IR expression on CD4 T cells. (**A**) CD4/CD279 MFI with hsCRP (*p* = 0.0393, N = 56); (**B**) CD4/CD279 MFI with ESR (*p* = 0.0096, N = 51); (**C**) CD4/CD279% with hsCRP (*p* = 0.0251, N = 56); (**D**) CD4/TIM3% with hsCRP (*p* = 0.0459, N = 56); (**E**) CD4/TIGIT MFI with hsCRP (*p* = 0.0156, N = 45); (**F**) CD4/CD127 MFI with DAS28-ESR (*p* = 0.0321, N = 40); (**G**) CD4/CD127 MFI with DAS28-CRP (*p* = 0.0145, N = 40).

**Figure 2 cells-13-00403-f002:**
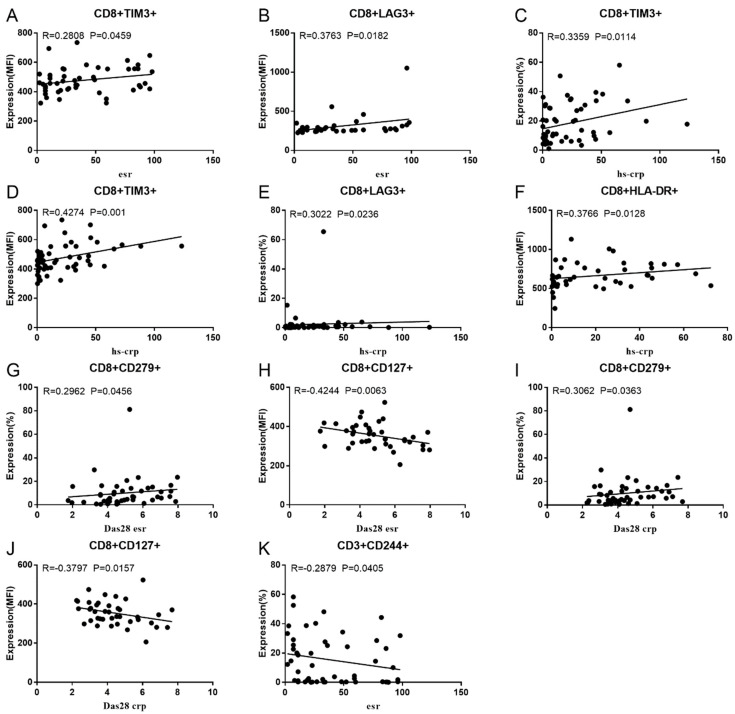
Significant correlation between RA disease activity and Co-IR expression on CD3 and CD8 T cells. (**A**) CD8/TIM3 MFI with ESR (*p* = 0.0459, N = 51); (**B**) CD8/LAG3 MFI with ESR (*p* = 0.0182, N = 39); (**C**) CD8/TIM3% with hsCRP (*p* = 0.0114, N = 56); (**D**) CD8/TIM3 MFI with hsCRP (*p* = 0.001, N = 56); (**E**) CD8/LAG3% with hsCRP (*p* = 0.0236, N = 56); (**F**) CD8+HLA-DR MFI with hsCRP (*p* = 0.0128, N = 43); (**G**) CD8/CD279% with DAS28-ESR (*p* = 0.0456, N = 46) (**H**); CD8/CD127 MFI with DAS28-ESR (*p* = 0.0063, N = 40); (**I**) (CD8/CD279% with DAS28-CRP (*p* = 0.0363, N = 47); (**J**) CD8/CD127 MFI with DAS28-CRP (*p* = 0.0157, N = 40); (**K**) CD3/CD244% with ESR (*p* = 0.0405, N = 51).

**Figure 3 cells-13-00403-f003:**
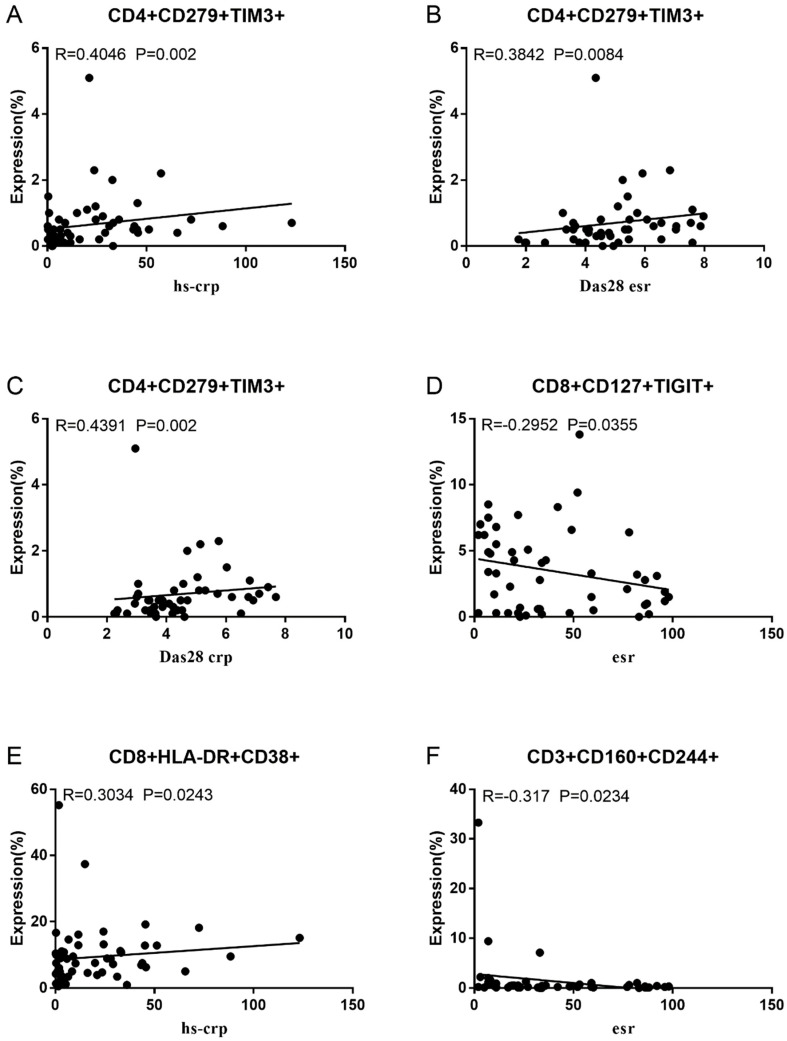
Significant correlation between RA disease activity and expression of two Co-IRs on T cells. (**A**) CD4+CD279+TIM3+% with hsCRP (*p* = 0.002, N = 56); (**B**) CD4+CD279+TIM3+% with DAS28-ESR (*p* = 0.0084, N = 46); (**C**) CD4+CD279+TIM3+% with DAS-28 CRP (*p* = 0.002, N = 47); (**D**) CD8+CD127+TIGIT+% with ESR (*p* = 0.0355, N = 51); (**E**) CD8+HLA-DR+CD38+% with hsCRP (*p* = 0.0243, N = 55); (**F**) CD3+CD160+CD244+% with ESR (*p* = 0.0234, N = 51).

**Figure 4 cells-13-00403-f004:**
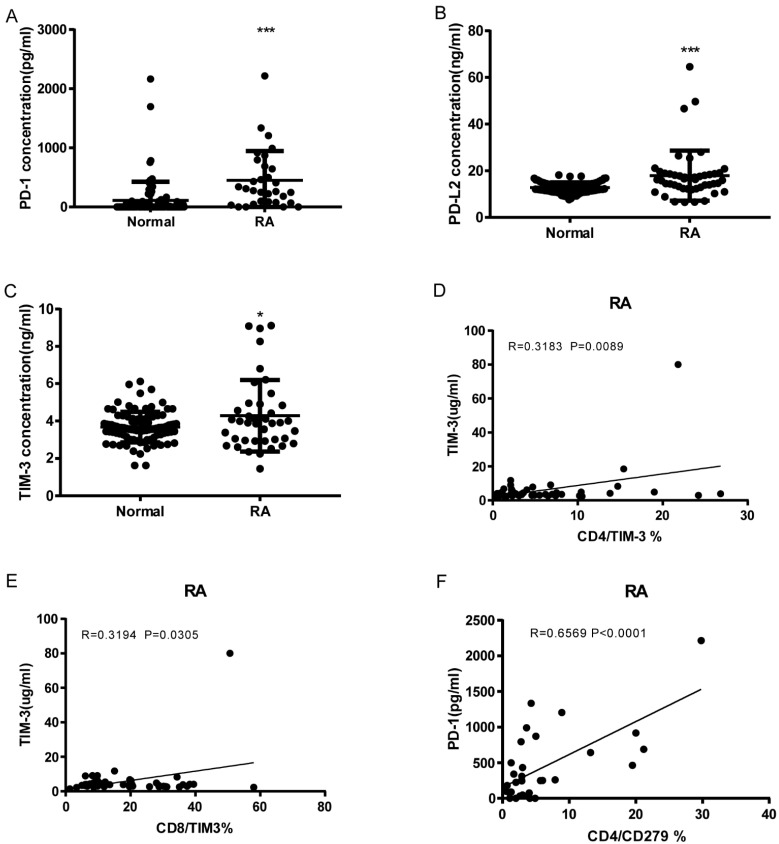
Soluble PD-1, PDL-2, and Tim3 levels elevated RA and correlation with T cell Co-IR expression. (**A**) sPD1: 450.5 ± 89.01, *n* = 31 vs. 111.1 ± 32.92, *n* = 92 *p* < 0.0001; (**B**) sPDL2: 17.9 ± 1.566, *n* = 47 vs. 12.78 ± 0.2314, *n* = 92 *p* < 0.0001; (**C**) sTIM3: 4.288 ± 0.3015, *n* = 40 vs. 3.675 ± 0.08441, *n* = 92 *p* = 0.0104); (**D**) sTIM3 levels were correlated with CD4+TIM3+% (*p* = 0.0089, N = 46); (**E**) CD8+TIM3+% (*p* = 0.0305, N = 46); (**F**) sPD1 levels were correlated with CD4+CD279+ % (*p* < 0.0001, N = 31). ***, *p* < 0.001; *, *p* < 0.05.

**Table 1 cells-13-00403-t001:** T cell expression of single Co-IRs in RA and Normal groups.

Markers	Mean ± SD (Number)		*p* Value
	RA	Normal	
CD4+CD279	4.6 (9.625) (N = 48)	3.1 (6.9) (N = 27)	0.0088
CD4+TIM3+	4.75 (5.75) (N = 48)	2.4 (2.225) (N = 26)	0.0171
CD4+CTLA4+	0.15 (0.3) (N = 46)	0.1 (0.1) (N = 27)	0.0522
CD4+TIGIT+	25.35 (11.23) (N = 48)	20.1 (5.73) (N = 26)	0.0016 *
CD8+CD279+	5.7 (8.3) (N = 47)	3.1 (6.2) (N = 27)	0.0253
CD8+TIM3+	12.9 (20.255) (N = 48)	7.5 (8.8) (N = 27)	0.0012 *
CD8+LAG3+	0.55 (1) (N = 46)	0.1 (0.3) (N = 27)	0.0015 *
CD8+CD127+	21.2 (22.63) (N = 48)	40.2 (36.2) (N = 27)	0.0012 *
CD8+TIGIT+	47.95 (23.65) (N = 48)	25.4 (12.9) (N = 27)	<0.0001 *
CD8+CD160+	36.7 (18.1) (N = 47)	28.6 (15.6) (N = 27)	0.0513
CD8+HLA-DR+	43.8 (24.8) (N = 31)	15.4 (13.85) (N = 17)	<0.0001 *
CD8+CD38+	26.9 (15.8) (N = 31)	23.9 (11.9) (N = 17)	0.2493
CD3+CD244+	8.65 (30.68) (N = 48)	0.45 (8.45) (N = 26)	0.0791
CD3+CD279+	4.7 (9.33) (N = 46)	2 (5.7) (N = 27)	0.025
CD3+TIGIT+	40 (31.8) (N = 47)	22.5 (18.6) (N = 27)	0.0001 *

* Bonferroni correction *p* = 0.05/21 = 0.00238 are significant for multiple comparison.

**Table 2 cells-13-00403-t002:** T cell expression of two Co-IRs in RA and Normal groups.

Markers	Mean ± SD (Number)		*p* Value
	RA	Normal	
CD4+CD279+TIM3+	0.4 (0.6) (N = 47)	0.2 (0.4) (N = 27)	0.0246
CD8+CD279+TIM3+	0.2 (0.325) (N = 46)	0.1 (0.1) (N = 26)	0.0052 *
CD8+CD127+TIGIT+	2.3 (4.3) (N = 47)	2 (3.5) (N = 27)	0.4903
CD8+CD160+CD244+	0.2 (0.4) (N = 47)	0.1 (0.55) (N = 25)	0.912
CD8+HLA-DR+CD127+	0.6 (0.4) (N = 19)	0.9 (2.3) (N = 19)	0.3567
CD8+HLA-DR+CD38+	7.5 (6.6) (N = 47)	3.1 (3.3) (N = 27)	0.0004 *
CD3+CD160+CD244+	0.2 (0.5) (N = 47)	0.2 (0.2) (N = 27)	0.1262
CD3+CD279+TIGIT+	3.05 (4.95) (N = 46)	1.7 (2.975) (N = 26)	0.0657

* Bonferroni correction *p* = 0.05/11 = 0.00455 are significant for multiple comparison.

## Data Availability

The data presented in this study are available on request from the corresponding author.

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
