# Peer review of "Comprehensive Co-Inhibitory Receptor (Co-IR) Expression on T Cells and Soluble Proteins in Rheumatoid Arthritis"

_cells, 2024, doi:10.3390/cells13050403_

Round 1

Reviewer 1 Report

Comments and Suggestions for Authors

Major comments

Please add plots of the gating strategy, at least to the supplementary data.

The information on the patient cohort needs to be more detailed, for example treatment information is absent and no information is provided on the healthy controls.

Adjustment for multiple comparisons is required given the number of t-tests and correlations that have been run for the work. As it currently stands there is a significant risk of false positives.

During the results, please ensure it is clear if you are discussing MFI differences or percentage differences as it is not always clear.

Minor comments

Please proofread the document again, the language is sometimes unclear.

On line 87 there seems to be a typing error regarding PBMCs/SFMCs.

Please add the flow cytometer used

Please define what is meant by the flow cytometry HIP method in line 106.

Comments on the Quality of English Language

Language is overall of a moderate standard but requires some proofreading/editing both for minor typing errors and to make understanding the text easier.

Author Response

Reviewer 1:

Major comments

Reviewer: Please add plots of the gating strategy, at least to the supplementary data.

Response: We appreciate suggestions. We include graphs of the gating strategy approach as supplementary figures.

Reviewer: The information on the patient cohort needs to be more detailed, for example treatment information is absent and no information is provided on the healthy controls.

Response: Thanks for the suggestion. We provide description of clinical x-ray stages, RF positive status, and therapy information for cohort patients on methods, and give thorough a summary for RA and healthy controls detail information, in Supplementary Table 1.

Reviewer: Adjustment for multiple comparisons is required given the number of t-tests and correlations that have been run for the work. As it currently stands there is a significant risk of false positives.

Response: Thanks for valuable comments. We also emphasize that multiple comparisons must be considered for p values with Bonferroni adjustment.

Reviewer: During the results, please ensure it is clear if you are discussing MFI differences or percentage differences as it is not always clear.

Response: Thanks for the suggestion. We goal to be more explicit when addressing percentage and MFI on statement. In addition, MFI in bold font distinguish it from percentage differences.

Minor comments

Reviewer: Please proofread the document again, the language is sometimes unclear.

Response: We thoroughly proofread the text and clarify the language.

Reviewer: On line 87 there seems to be a typing error regarding PBMCs/SFMCs.

Response: We removed the typographical error with SFMCs.

Reviewer: Please add the flow cytometer used

Response: We use the FACSCanto II flow cytometer.

Reviewer: Please define what is meant by the flow cytometry HIP method in line 106.

Response: We added the definition of cytometry HIP technique. The phenotype of immune cell subsets was determined utilizing a full four-color flow cytometric analysis adopting the HIP method of standardized immunophenotyping procedures, as well as a detailed understanding of the parameters of healthy vs diseased or perturbed human immune systems.

Reference 7: Maecker HT, McCoy JP, Nussenblatt R. Standardizing immunophenotyping for the Human Immunology Project. Nat Rev Immunol 2012;12:191-200.

Reviewer 2 Report

Comments and Suggestions for Authors

Intersting article.

The sample size was appropriate.

Patients were treated with conventional D-MARDs. could this treatment have interfered with the results? Were any patients treated by biologics or targeted synthetic D-MARDs? how long was the delay between immunophenotypic analyses and D-MARD initiation?

The serum protein levels of CO-IRs and PD1, PD2 and Tim3 were assessed using ELISA technique. The gene expression could also have been performed using Q-RT-PCR technique.

the pathophysiological roles of molecules and receptors / markers such as TIGIT, CD127, MFI, LAG-3, TIM3, CD279, CD127, CD244 should be indicated more precisely (not only in the discussion section), in Table 1 or in the text for instance . Are they all CO-IRs?

Functions of PD-1 and PD-2 should also be more precisely indicated.

Additionnal functional tests (effects on T cell proliferation and cytokine expression) could have been performed on cell cultures.

Could the dosage of these molecules be performed routinely to predict RA activity? What could be the usefulness / relevance of these molecular markers in clinical practice?

Comments on the Quality of English Language

Intersting article.

The sample size was appropriate.

Patients were treated with conventional D-MARDs. could this treatment have interfered with the results? Were any patients treated by biologics or targeted synthetic D-MARDs? how long was the delay between immunophenotypic analyses and D-MARD initiation?

The serum protein levels of CO-IRs and PD1, PD2 and Tim3 were assessed using ELISA technique. The gene expression could also have been performed using Q-RT-PCR technique.

the pathophysiological roles of molecules and receptors / markers such as TIGIT, CD127, MFI, LAG-3, TIM3, CD279, CD127, CD244 should be indicated more precisely (not only in the discussion section), in Table 1 or in the text for instance . Are they all CO-IRs?

Functions of PD-1 and PD-2 should also be more precisely indicated.

Additionnal functional tests (effects on T cell proliferation and cytokine expression) could have been performed on cell cultures.

Could the dosage of these molecules be performed routinely to predict RA activity? What could be the usefulness / relevance of these molecular markers in clinical practice?

Author Response

Reviewer 2

Reviewer: Patients were treated with conventional D-MARDs. could this treatment have interfered with the results? Were any patients treated by biologics or targeted synthetic D-MARDs? how long was the delay between immunophenotypic analyses and DMARD initiation?

Response: Thanks for Valuable comments. Biologics or targeted synthetic D-MARDs and  four key conventional D-MARD therapeutic including MTX, HCQ, Sulfasalazine and Leflunomide were presented in supplementary Table 1. We are unable to determine the exact duration of the delay between immunophenotypic analysis and DMARD initiation in each patient. Rather, we include details about the RF status and X-ray stages in the supplemental table.

Reviewer: The serum protein levels of CO-IRs and PD1, PD2 and Tim3 were assessed using ELISA technique. The gene expression could also have been performed using Q-RT-PCR technique.

Response: We appreciate for valuable suggestions. This study focuses on protein relationships with RA activity and status. Gene expression employing Q-RT-PCR was not done. We will use RNA sequencing to explore CO-IR alterations in a prospective RA investigation after initiating biologics or targeted synthetic D-MARDs.

Reviewer: the pathophysiological roles of molecules and receptors / markers such as TIGIT, CD127, MFI, LAG-3, TIM3, CD279, CD127, CD244 should be indicated more precisely (not only in the discussion section), in Table 1 or in the text for instance . Are they all CO-IRs?

Response: We appreciate for valuable suggestions. TIGIT, LAG-3, TIM3, CD279, and PD-1 significant CO-IRs play pathophysiological roles in RA, as explained in the introduction. CD127 is required for T cell proliferation, whereas CD244 delivers stimulatory or inhibitory signals.

Reviewer: Functions of PD-1 and PD-L2 should also be more precisely indicated.

Response: The functional and pathophysiological roles of PD-1 and PD-2 are more precisely indicated. Introduction: Binding to PD-1, PD-L1, or PD-L2 promotes subsequent PD-1-linked transduction events. This can impair TCR signaling via feedback interference and diminish antiapoptotic proteins, thereby limiting T-cell survival, proliferation, and immunological function. An immune checkpoint modulates immune activation and self-tolerance.

Discussion: TIM3 pathways and ligands that connect PD-1 and PD-L1/2 might mitigate RA autoimmunity. sTIM3 and sPD-1 can compete with the ligand, thereby blocking the inhibiting nature of membrane-bound PD-1/TIM-3 signals and facilitating T-cell exertion . sPD-1 accelerates the onset of collagen-induced arthritis through Th1 and Th17 pathways

Reviewer: Additionnal functional tests (effects on T cell proliferation and cytokine expression) could have been performed on cell cultures.

Response: We appreciate your important suggestions. In our discussion, we pointed out that the fundamental limitation of our work was the inability to clearly identify T cell immunophenotype contributions to the etiology of RA. In the future, single-cell RNA sequencing (scRNA-seq) could help resolve specific immunophenotypes follow by evaluating T cell proliferation and cytokine production in cell cultures. 

Reviewer: Could the dosage of these molecules be performed routinely to predict RA activity? What could be the usefulness / relevance of these molecular markers in clinical practice?

Response: We thanks to your thoughtful suggestions. Co-IR molecules co-localize with activation markers that are expressed on T cells. The comprehensive determination of the dynamic balance of T cells with these molecules may guide the selection of biologics or targeted synthetic D-MARDs and the therapy of rheumatoid arthritis. 

Reviewer 3 Report

Comments and Suggestions for Authors

The authors present a study on the role of co-inhibitory receptors expressed on T cells and soluble proteins in both the pathogenesis of rheumatoid arthritis and the prediction of disease activity.  

1) In the introduction, the authors should better describe what co-inhibitory receptors are and their mechanism of action in immune-mediated diseases.
2) In the description of the patient population, the number of patients studied should be specified.
3) The criteria used for the diagnosis of rheumatoid arthritis need references to the literature.
4) Authors should specify which conventional DMARDs were used to treat patients.
5) Authors should specify that the study population is normally distributed and thus justify the use of mean and standard deviation. Otherwise, it would be more appropriate to describe the results as median and interquartile range.  Nonparametric tests would be more appropriate for comparing relatively small patient populations.
6) In the discussion, the term CD4/CD8 T cells may be confused with double-positive T cells. Better to write CD4+ and CD8+ T cells.
7) It should be specified that the CTLA-4 Ig fusion molecule blocks co-stimulation of T cells.
8) The authors should cite doi: 10.1056/NEJMoMoa2209856 as the monoclonal antibody peresolimab that stimulates the PD-1 pathway for rheumatoid arthritis therapy was tested in this clinical trial.
8) The occurrence of immune-related adverse events during cancer therapy with immune check-point inhibitors should also be discussed.
9) The study was conducted on lymphocytes isolated from peripheral blood. The authors should discuss the significance of their results in light of the fact that the cells responsible for rheumatoid arthritis are present in the synovium rather than in the peripheral blood, where they exert their inflammatory activity. For this reason, currently many studies are conducted on T cells isolated from synovial biopsy specimens.

Comments on the Quality of English Language

The English is grammatically correct, although the style is sometimes a bit verbose and difficult to understand.

Author Response

Reviewer 3: The authors present a study on the role of co-inhibitory receptors expressed on T cells and soluble proteins in both the pathogenesis of rheumatoid arthritis and the prediction of disease activity.  

Reviewer: In the introduction, the authors should better describe what co-inhibitory receptors are and their mechanism of action in immune-mediated diseases.

Response: We thanks to your thoughtful suggestions. In the introduction, we explore co-inhibitory receptors' negative functions and mechanisms of action in immune-mediated illnesses.

Reviewer: In the description of the patient population, the number of patients studied should be specified.

Response: We thanks to your thoughtful suggestions. We specified the number of 48 RA patients, with comprehensive information in the method section and Supplemental Table 1.

Reviewer: The criteria used for the diagnosis of rheumatoid arthritis need references to the literature.

Response: We thanks to your thoughtful suggestions. We cited the reference of 2010 rheumatoid arthritis classification criteria: an American College of Rheumatology/European League Against Rheumatism collaborative initiative used for the diagnosis of rheumatoid arthritis

Reviewer: Authors should specify which conventional DMARDs were used to treat patients.

Response: We thanks to your thoughtful suggestions. We specified the conventional DMARDs use the method section and Supplemental Table 1.

Reviewer: Authors should specify that the study since normal the use of mean and standard deviation. Otherwise, it would be more appropriate to describe the results as median and interquartile range.  Nonparametric tests would be more appropriate for comparing relatively small patient population

Response: We thanks to your thoughtful suggestions. We performed the nonparametric tests and provide fresh tables with median and interquartile ranges for a small patient sample sizes as a suggestion. 

Reviewer: In the discussion, the term CD4/CD8 T cells may be confused with double-positive T cells. Better to write CD4+ and CD8+ T cells.

Response: We corrected CD4/CD8 T cells to CD4+ and CD8+ T cells

Reviewer: It should be specified that the CTLA-4 Ig fusion molecule blocks co-stimulation of T cells.

Response: We thanks to your thoughtful suggestions. We specified that the CTLA-4 Ig fusion molecule blocks co-stimulation of T cells.

Reviewer: The authors should cite doi: 10.1056/NEJMoMoa2209856 as the monoclonal antibody peresolimab that stimulates the PD-1 pathway for rheumatoid arthritis therapy was tested in this clinical trial.

Response: We thanks to your thoughtful suggestions. We quote the phase 2a trial in which peresolimab showed efficacy in patients with RA. The reduction from baseline in the DAS28-CRP was significantly greater in the 700-mg peresolimab group.

Reviewer: The occurrence of immune-related adverse events during cancer therapy with immune check-point inhibitors should also be discussed.

Response: We thanks to your thoughtful suggestions. We add the immune-related adverse events during cancer therapy with immune check-point inhibitors in the discussion.

Reviewer: The study was conducted on lymphocytes isolated from peripheral blood. The authors should discuss the significance of their results in light of the fact that the cells responsible for rheumatoid arthritis are present in the synovium rather than in the peripheral blood, where they exert their inflammatory activity. For this reason, currently many studies are conducted on T cells isolated from synovial biopsy specimens.

Response: We thanks to your thoughtful suggestions. We add concern of “The synovium is the epicenter of rheumatoid arthritis-causing cells, not the peripheral circulation, where they function to initiate the inflammatory response. Consequently, an extensive array of investigations are currently underway pertaining to T cells obtained from synovial biopsy samples”.

Round 2

Reviewer 3 Report

Comments and Suggestions for Authors

The authors modified the manuscript in accordance with the suggestions and answered the questions thoroughly. In my opinion, the manuscript now deserves to be published in the journal Cells

Author Response

Thanksfor positive comments

Ji Yih Chen  MD